# Effects of a Novel Proprioceptive Rehabilitation Device on Shoulder Joint Position Sense, Pain and Function

**DOI:** 10.3390/medicina58091248

**Published:** 2022-09-09

**Authors:** Alexandra Camelia Gliga, Nicolae Emilian Neagu, Septimiu Voidazan, Horatiu Valeriu Popoviciu, Tiberiu Bataga

**Affiliations:** 1Institution Organizing University Doctoral Studies of George Emil Palade University of Medicine, Pharmacy, Science, and Technology of Targu Mures, 540139 Targu Mures, Romania; 2Laboratory of Rehabilitation, Physical Medicine and Balneology, County Clinical Emergency Hospital of Targu Mures, 540136 Targu Mures, Romania; 3Department of Functional and Complementary Sciences, Faculty of Medicine, George Emil Palade University of Medicine, Pharmacy, Science, and Technology of Targu Mures, 540139 Targu Mures, Romania; 4Department of Clinical Sciences and Internal Medicine, Faculty of Medicine, George Emil Palade University of Medicine, Pharmacy, Science, and Technology of Targu Mures, 540139 Targu Mures, Romania; 5Clinic of Rheumatology, County Clinical Emergency Hospital of Targu Mures, 540136 Targu Mures, Romania; 6Department of Clinical Sciences, Faculty of Medicine, George Emil Palade University of Medicine, Pharmacy, Science, and Technology of Targu Mures, 540139 Targu Mures, Romania; 7Clinic of Orthopaedics and Traumatology, County Clinical Emergency Hospital of Targu Mures, 540136 Targu Mures, Romania

**Keywords:** rehabilitation, device, shoulder joint, proprioception, joint position sense, DASH

## Abstract

*Background and Objectives*: Shoulder disorders are associated with pain, restricted range of motion and muscular strength, moderate disability and diminished proprioception. This study aimed to compare the effectiveness of an innovative technology-supported and a classical therapist-based proprioceptive training program in addition to conventional physiotherapy, on joint position sense (JPS), pain and function, in individuals with different musculoskeletal shoulder disorders, such as rotator cuff tear, subacromial impingement syndrome and superior labrum anterior and posterior tear. The innovative element of the proprioceptive training programme consists of the use of the Kinesimeter, a device created for both training and assessing shoulder JPS. *Materials and Methods*: The shoulder JPS test and the DASH outcome questionnaire were applied to fifty-five individuals (28 females, 27 males, mean age 56.31 ± 6.75), divided into three groups: 17 in the conventional physiotherapy group (control group); 19 in the conventional physiotherapy + classical proprioceptive training program group (CPT group); and 19 in the conventional physiotherapy + innovative proprioceptive training program group (KPT group). Assessments were performed before and after a four-week rehabilitation program, with five physiotherapy sessions per week. *Results*: When baseline and post-intervention results were compared, the value of the shoulder JPS and DASH outcome questionnaire improved significantly for the KPT and CPT groups (all *p* < 0.001). Both KPT and CPT groups showed statistically significant improvements in JPS, pain and function, compared to the control group which received no proprioceptive training (all *p* < 0.05). However, the KPT group showed no significant benefits compared to the CPT group. *Conclusions*: Our findings indicate that using the Kinesimeter device as a novel, innovative proprioceptive training tool has similar effects as the classical proprioceptive training programs among individuals with different non-operated musculoskeletal shoulder disorders such as: rotator cuff tear, subacromial impingement syndrome, and superior labrum anterior and posterior tear.

## 1. Introduction

The term “proprioception” is often found in the literature, being formulated in various definitions, varying depending on different concepts and areas of applicability. Proprioception refers to the awareness of the position and movements of the limbs, torso and head [1]. Often considered the sixth sense, proprioception is sensory information about the direction of movement, location in space, and speed, as well as the muscle activation transmitted to the central nervous system [2]. The proprioceptive sense is the source of information received from receptors in muscles, tendons and joints. These receptors are mechanically and baro-sensitive [3].

Proprioception influences the accuracy and precision of the movement [4]. Kelso et al. (1980) demonstrated that, following the replacement of the articular capsule, the subjects encountered slightly affected accuracy of the spatial position of the respective segment. In contrast, the accuracy of the movements of the distal segments was severely disturbed [5]. The onset of motor command is also influenced by proprioceptive feedback [6].

Numerous studies have shown correlations between deficits in proprioceptive motor control and musculoskeletal shoulder injuries [7,8,9]. During daily activities, the glenohumeral joint performs a wide range of movements, at various amplitudes and speeds, thus being subjected to a series of translational forces [10]. The proper shoulder functioning involves synchronous and harmonious movements at five levels: scapulohumeral, scapulothoracic, subacromial, acromioclavicular and sternoclavicular.

Shoulder disorders occur either traumatic, due to external factors, following an injury, fall, etc., or non-traumatic, due to degenerative or other internal factors. Current research shows that proprioception is severely affected by rotator cuff (RC) injuries, superior labrum anterior and posterior (SLAP) injuries, and subacromial impingement syndrome (SIS) [8,9,11]. RC injuries can also occur as a result of severe trauma, but in most cases, they result from tendon degeneration caused by SIS [12]. Although RC injuries have also been reported in children, they are characteristic of people over forty. Up to 30–70% of patients with shoulder pain have an associated RC injury [13]. In the case of this pathology, the maximum pain occurs during the flexion or abduction movement, between 90–120°. Patients often report the presence of intra-articular crepitus and night pain [14]. 

The occurrence of impingement in the subacromial bursa is common, this being the cause of shoulder pain in 44–65% of shoulder pathologies [15]. Overusing the shoulder or imperfect centring of the humeral head in the glenoid are the main cause of inflammation and finally degeneration [16]. The association of SIS with RC injuries is common. The symptoms include pain, decreased muscle strength, crepitus, etc. [14].

Injuries to the glenoid labrum may occur due to degenerative alterations, but also after traumatic injuries, resulting from dislocations or subluxations of the glenohumeral joint. SLAP lesions involve a rupture of the superior glenoid labrum, with or without the involvement of the long head tendon of the brachial biceps. Repetitive overhead movements most often cause the production mechanism. The main symptom is pain, accompanied by decreased muscle strength. Patients often describe the pain as located “deep inside” the joint. SLAP injuries can cause glenohumeral instability and RC tears.

In the case of these three shoulder disorders, conservative treatment involves immobilization, pharmaceutical treatment, physical therapy, electrotherapy, thermotherapy, kinesiology tapping, etc. [17,18]. Patients who do not have a favourable response to conservative treatment will be referred for surgery, with the primary goal of reducing pain, improving range of motion, muscle strength and restoring shoulder function [19].

After a shoulder injury, physiotherapy will start as soon as possible, taking into account the indications of the multidisciplinary team, the pathology, the type of treatment applied (surgical or conservative), and the needs and characteristics of the patient. The ultimate goal of rehabilitation will be to regain professionalism and return to the previous socio-professional life [20]. The stages of the rehabilitation programs in the orthopaedic and traumatic disorders of the shoulder include five main phases, adapted according to pathology, objectives and individual: (1) stage of immobilization after the traumatism or after the surgery; (2) stage of mechanical harmonization of the shoulder joint (includes manual therapy, Codman-type exercises, isometric exercises, passive shoulder mobilizations, etc.); (3) stage of complete rehabilitation of the range of motion (includes stretching, manual therapy, assisted or free active shoulder mobilizations, etc.); (4) stage of increasing muscle strength and improving motor control (includes exercises with progressive resistance, in open or closed kinetic chain, exercises for improving motor control and proprioception, etc.); and (5) stage of regaining professionalism (includes exercise training, plyometric and coordination exercises) [21,22,23,24]. Assuming that no two patients are identical, it is essential to understand that these stages have the prominent role of guiding rehabilitation programs and creating pyramid-shaped progress [25]. 

As previously presented, complete rehabilitation of the affected proprioceptive function is the subject of the fourth stage of the program. During this stage, different means and methods are used, such as neuromuscular facilitation techniques (mainly rhythmic stabilization, elective in the rehabilitation of proprioception), active mobilizations with manual resistance, followed by different external resistances (weights, elastic bands, dumbbells, or water resistance—in case of hydrotherapy) [26]. Different unstable rehabilitation tools are beneficial for enhancing reflex responses, somatosensory sense and proprioception [27]. The advancement of technology has helped physiotherapists, creating various robots and exoskeletons to facilitate proprioception and improve motor control [28,29]. One of the benefits of these technologies is the reproduction of real-life scenarios, creating a safe and individualized environment for the patient, with the help of which to arouse his interest, voluntarily involving him [30].

This study aimed to compare the effectiveness of an innovative technology-supported and a classical therapist-based proprioceptive training program in addition to conventional physiotherapy, on joint position sense, pain and performing ability, work ability and sports and art activity ability in individuals with different musculoskeletal shoulder disorders, such as RC tear, SIS and SLAP tear. We hypothesized that: (1) the use of the Kinesimeter—a novel proprioceptive rehabilitation device—would significantly improve shoulder JPS, pain and function; and (2) the results obtained by the innovative technology-supported proprioceptive training group would be superior to the ones obtained by the classical therapist-based proprioceptive training group and the ones obtained by the conventional physiotherapy group.

## 2. Materials and Methods

### 2.1. Study Design and Participants

In this prospective, single-centre, randomized trial, we included fifty-five participants who received four weeks of shoulder joint rehabilitation in the Laboratory of Rehabilitation, Physical Medicine and Balneology from the County Clinical Emergency Hospital of Targu Mures, Romania. This study was conducted between October 2020 and October 2021 on participants who voluntarily agreed to be part of our research. The aim and the protocol of this research were explained to all participants, and an informed consent form was signed before the inclusion. Moreover, two ethical committee approvals were received, one from the County Clinical Emergency Hospital of Targu Mures (Approval number 1149 from 15 October 2020) and one from the “George Emil Palade” University of Medicine, Pharmacy, Science and Technology of Targu Mures (Approval number Ad. 23666 from 16 October 2020).

For all participants, the inclusion criteria were: (1) clinical diagnoses of all types of RC tears, based on Collin’s classification, first or second stage of SIS, based on Neer’s classification, and first stage SLAP tear, based on Snyder’s classification; (2) ability to perform painless active range of motion mobilizations of the shoulder with minimum values of 100° flexion and abduction, 35° extension, 40° internal and external rotations; (3) localization of the shoulder disorder in the dominant upper limb; (4) age 45 to 65 years; and (5) moderate physical activity level. Exclusion criteria: (1) history of recent surgery of the dominant upper limb; (2) history of neurological, severe cardiovascular, psychiatric, diabetes or infectious conditions; and (3) undergoing pharmaceutical treatment that might affect the outcome results. It was explained to the participants that they had to inform the investigators about changing their level of physical activity or starting to use drug treatments and that they have the right to withdraw their participation from our research at any time.

Based on the type of rehabilitation program, with the use of digital software participants were divided into three groups that had a similar number of subjects and anthropometric characteristics: conventional physiotherapy (control group; *n* = 17 participants), conventional physiotherapy + classical therapist-based proprioceptive training program (CPT group; *n* = 19 participants) and conventional physiotherapy + innovative technology-supported proprioceptive training with the use of the Kinesimeter device (KPT group; *n* = 19 participants).

### 2.2. Rehabilitation Programs

For four weeks, all the participants benefited from a five-day-a-week program of individualized and personalized medical rehabilitation. Each session duration and the main parameters of effort were similar for all three groups. All participants started the shoulder rehabilitation session with approximately 20 min of warm-up, followed by 25 min of exercises specific to each study group and finalized with approximately 5 min of stretching and relaxation. 

The exercises specific to the control group consisted of assisted or free active mobilizations of the upper limb. The CPT group underwent a specific active proprioceptive training program. For this, we used: (1) neuromuscular facilitation techniques; (2) active exercises, performed on various unstable surfaces, with or without external resistances, such as weights and elastic bands; (3) somatosensory stimulation training with vibration therapy; and (4) shoulder plyometric program, designed by Boston Sports Medicine and Research Institute [31]. All the exercises were performed with both upper limbs, with or without visual control.

The KPT group underwent an innovative technology-supported proprioceptive training program, using the Kinesimeter device. Designed within the Department of Functional and Complementary Sciences, Faculty of Medicine, “George Emil Palade” University of Medicine, Pharmacy, Science, and Technology of Targu Mures, Romania, the Kinesimeter is a precise device which can assess or train a 2D movement of the upper limb. The hardware of the device consists of a fixed vertical stand and a movable horizontal arm, presented in Figure 1a. After connecting the Kinesimeter to a laptop, a software interface reproduces the movement of the upper limb, creating real-time oscillograms and providing data related to the shape, amplitude, frequency, duration and chronology, Figure 1b. Thus, in the case of proprioceptive training with visual control, the participant had high accuracy feedback. In the motor suggestibility training, without visual control, the Kinesimeter provided auditory feedback of varying intensity when the reference value was exceeded or not reached, Figure 1c. Also, different weights were attached to the arm of the device, doing possible proprioceptive training with different loads. The innovative technology-supported proprioceptive training program included exercises from seated or lying positions, with and without visual control, attached weights or auditory feedback. All the exercises were performed with both upper limbs.

### 2.3. Outcome Measures

The shoulder JPS assessment and the DASH Outcome Questionnaire were applied at the beginning and at the end of the rehabilitation program.

For shoulder JPS assessment we used the same device as for the innovative proprioceptive training applied to the KPT group. The description, validity and reliability of the Kinesimeter, used as an assessment device for precise angles and JPS evaluation, have been demonstrated in two previous studies [32,33]. 

The assessment technique used in our study was passively positioning the upper limb at the same target angle for all participants (flexion and abduction—60°; internal and external rotation—35°; and extension—25°), maintaining the position for no longer than 10 s, returning to the initial relaxed position and actively reproducing the previous movement toward the perceived angle. Spinger et al. (2017) demonstrated similar levels of reposition accuracy when the subject had 3 to 12 s to memorize the target angle, [34]. Still, for not allowing fatigue to influence the results, most of the studies allow the participants to focus on the reference angle for no longer than 10 s [35,36]. Three trials of assessing the error of active reproduction (difference between the target angle and the perceived angle) were carried out, for each shoulder movement. The mean of all three trials was recorded as the absolute error.

For accurate results, assessments were done without visual control and all participants wore eye masks. JPS assessments of flexion, extension and abduction were performed while the participant was seated, with anatomically positioned upper limb, extended elbow and supinated hand. JPS assessments of internal and external rotations were performed while the participant was lying, with 90° abducted upper limb, 90° flexed elbow and neutrally positioned hand.

The Disability of the Arm, Shoulder, and Hand Questionnaire (American Academy of Orthopaedic Surgeons, Upper Extremity Collaborative Group, Rosemont, IL, USA) was used to assess (1) pain and performing ability (30 items); (2) work and ability (4 items); and (3) sports and activity ability (4 items). The Romanian official translation, developed by Oxford Outcomes Ltd., Oxford, UK was used for our research. All the 38 items were scored on a scale from 1 to 5, meaning: (1) no difficulty; (2) slightly difficult; (3) moderate difficult; (4) very difficult; and (5) not at all, [37]. After the questionnaire was applied, we used the following formulas to calculate the total score:Pain and performing ability score = (sum of scores answered − 1) ÷ number of questions answered × 25(1)
Work and ability score = [(sum of scores answered ÷ 4) − 1] × 25(2)
Sports and activity ability score = [(sum of scores answered ÷ 4) − 1] × 25(3)

For the interpretation of the values obtained, it is taken into account that a small DASH value represents a good functionality of the shoulder and a low level of pain [38].

### 2.4. Statistical Analysis

Graph Pad Prism for Windows (trial version), San Diego, CA, USA, was used for statistical calculations. The Shapiro-Wilk test was applied to assess the normality of continuous variables (i.e., age, height, weight and BMI). The student’s *t*-test was used to assess the differences between means of continuous variables (expressed as mean ± SD). The differences among constant variables of the three groups were investigated using ANOVA tests, an analysis appropriate for more than two groups. Using Bonferroni multiple comparison tests, we found the groups between whom there were statistically significant differences. We interpreted all tests against a *p* = 0.05 significance threshold and statistical significance was considered for *p*-values below the significance threshold.

## 3. Results

The study group consisted of fifty-five participants, (50.91% (*n* = 28) females and 49.09% (*n* = 27) males), with a mean age of 56.31 years (SD 6.75). The demographic characteristics of the participants and the statistical difference between the groups are summarized in Table 1.

Out of all the participants, 21.82% (*n* = 12) presented a recent history of SLAP tear, 40% (*n* = 22) a recent history of SIS and 38.18% (*n* = 21) a recent history of RC tear. All data regarding clinical information, sex and manual dominance are detailed in Table 2.

### 3.1. Joint Position Sense

In both CPT and KPT groups, the post-intervention absolute errors were lower than the baseline absolute errors for all studied movements (flexion, extension, internal rotation, external rotation and abduction) and the JPS was statistically significantly improved: all *p* < 0.001. In the control group, the post-intervention absolute errors were not lower than the baseline assessment absolute errors for extension (*p* = 0.275), internal (*p* = 0.122) and external (*p* = 0.203) rotations, but statistically significant lower, thus improved, for flexion (*p* = 0.047) and abduction (*p* = 0.027), Table 3.

Both KPT and CPT groups showed statistically significant improvements in JPS after four weeks of rehabilitation, compared to the control group which received no proprioceptive training (all *p* < 0.05). The comparative analysis with the control group finds the most considerable differences and thus the most significant improvements during flexion (*p* = 0.001), abduction (*p* < 0.001) and internal rotation (*p* < 0.001) movements for the KPT group, and during flexion (*p* = 0.001) and internal rotation (*p* = 0.001) movements for the CPT group. However, the recorded results reveal that there are no statistically significant differences in JPS improvements between the group that underwent a classical proprioceptive training program and the group that underwent an innovative proprioceptive training program, for all the studied categories (flexion *p* = 0.406; extension *p* = 0.57; abduction *p* = 0.173; internal rotation *p* = 0.537 and external rotation *p* = 0.688). The JPS inter-group comparative analysis is summarized in Figure 2.

### 3.2. DASH Outcome Questionnaire

In all three groups the post-intervention results were lower than the baseline assessment results and the scores for pain and performing ability, work ability, sports and activities ability were statistically significantly improved: all *p* < 0.05, Table 4.

Both KPT and CPT groups showed statistically significant improvements in the DASH Outcome Questionnaire score after four weeks of rehabilitation, compared to the control group which received no proprioceptive training (all *p* < 0.01). However, the recorded results reveal that there are no statistically significant differences in the pain and function of the shoulder between the group that underwent a classical proprioceptive training program and the group that underwent an innovative proprioceptive training program, for all the studied categories (pain and performing ability *p* = 0.063; work ability *p* = 0.099; and sports and activities *p* = 0.356). The DASH Outcome Questionnaire score inter-group comparative analysis is summarized in Figure 3. 

## 4. Discussion

This study aimed to compare the effectiveness of an innovative technology-supported and a classical therapist-based proprioceptive training program in addition to conventional physiotherapy on joint position sense, pain and function, in individuals with different musculoskeletal shoulder disorders, such as RC tear, SIS and SLAP tear. It was found that: (1) after four weeks of shoulder rehabilitation JPS improved significantly for CPT and KPT groups (all positions assessed *p* < 0.001) and only for the flexion and abduction position for the control group (*p* < 0.05); (2) when compared to the control group, both CPT and KPT groups showed superior improvements in the outcomes of shoulder JPS (all *p* > 0.05); (3) CPT or KPT groups were not superior to each other in improving shoulder JPS (all *p* > 0.05); (4) after treatment, significant improvements in the outcomes of DASH questionnaire were observed in all groups (all *p* < 0.05); (5) when compared to the control group, both CPT and KPT groups showed superior improvements in the outcomes of DASH questionnaire (all *p* < 0.01); (6) CPT or KPT groups were not superior to each other in improving pain and performing score, work ability score or sports and activities score (all *p* > 0.05). Our hypothesis that the use of the innovative Kinesimeter device would significantly improve shoulder JPS, pain and function is confirmed, while the hypothesis that the results obtained by the innovative technology-supported proprioceptive training group would be superior to the ones obtained by the classical therapist-based proprioceptive training group and the ones obtained by the conventional physiotherapy group is half denied. KPT group was not superior to CPT group in improving shoulder JPS, pain and function, but significantly superior to the control group.

In the current literature, many researchers are investigating the effects of different devices and robots used in upper limb rehabilitation [39,40]. Technology-supported rehabilitation was previously encountered in neurological pathologies, but due to the effects on reducing motor impairments and improving upper limb function, this therapeutical method is also highly recommended in orthopaedic disorders [41,42]. Besides the beneficial effects on enhancing rehabilitation progress, most rehabilitation devices also provide an objective assessment of the biomechanical data [43]. In previous studies, we proved the Kinesimeter device’s effectiveness in assessing precise angles and degrees of the shoulder’s range of motion and joint position sense [32,33]. In this study, we showed that, when used as a proprioceptive training device, the Kinesimeter can offer feedback to individuals, through a sound system (during audio-control training) or real-time oscillograms (during visual control training). As presented in Zhang et al. systematic review and meta-analysis, sensorimotor feedback can positively influence the training outcome [39].

Even though the KPT group benefited from accurate visual and audio feedback, the JPS post-intervention results were not sufficiently improved to exceed the results of the CPT group. Using a single device for a whole four-week JPS rehabilitation program was not superior to a classical combination program of different means and methods. Two features of the Kinesimeter that we believe negatively influenced the proprioceptive training results of the KPT group were: (1) performing all the JPS training exercises from seated or lying positions; and (2) performing only 2D movements of the upper limb. In addition to these, the results of the KPT group can also be caused by the lack of somatosensory stimulation training with vibration and by the impossibility of performing plyometric exercises during Kinesimeter use. Last but not least, the lack of use of PNF techniques in the case of the KPT group, whose role in the JPS rehabilitation is scientifically proven by many researchers, is another reason why our hypothesis was disproved, [44]. Our final results emphasize the important role of the physiotherapist in medical rehabilitation, a role that devices and technology cannot completely replace.

As shown before, combination therapies are more effective during the rehabilitation process than individual treatment techniques or monotherapies [45]. Some researchers suggest that the best results are achieved when technology-supported training is combined with conventional therapist-based rehabilitation treatment [46]. For this, we firmly believe that, due to the interactive, intense, repetitive training, the Kinesimeter can be a valuable complementary tool in the proprioceptive training of individuals with different shoulder disorders. 

### Limitations

The present study has several limitations. First, we did not investigate the correlation between the improvement of the JPS value and the improvement of the DASH outcome questionnaire score. Second, we did not investigate to what extent the effects of our interventions will continue over time, due to our final assessment applied at the end of the four-week rehabilitation program. In the future, we aim to individually investigate the differences between the improvements in the shoulder joint position sense, pain and function during the rehabilitation programs of the three musculoskeletal disorders presented in this study. More studies are needed to show the important role of proprioceptive rehabilitation in shoulder pathologies.

## 5. Conclusions

In this study, performing additional innovative technology-supported and classical therapist-based proprioceptive training like conventional physiotherapy caused significant improvements in shoulder joint position sense, pain and function. Our findings indicate that the use of the Kinesimeter device, as a novel innovative proprioceptive training tool, has similar results to the classical therapist-based proprioceptive training programs, among individuals with different non-operated musculoskeletal shoulder disorders such as RC tear, SIS and SLAP tear.

## Figures and Tables

**Figure 1 medicina-58-01248-f001:**
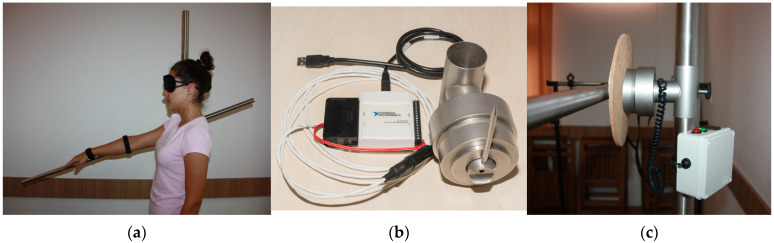
Presentation of the Kinesimeter device: (**a**) the Kinesimeter axis, fixed and mobile arm position for shoulder JPS rehabilitation in flexion; (**b**) measurement and data transfer mechanism for real-time oscillograms; (**c**) sound generator used for auditory feedback.

**Figure 2 medicina-58-01248-f002:**
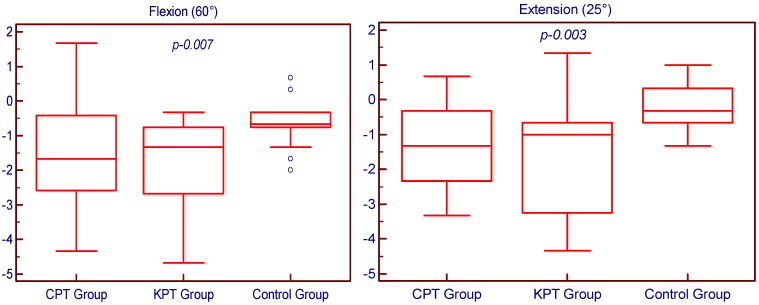
Inter–group comparative analysis of the shoulder JPS.

**Figure 3 medicina-58-01248-f003:**
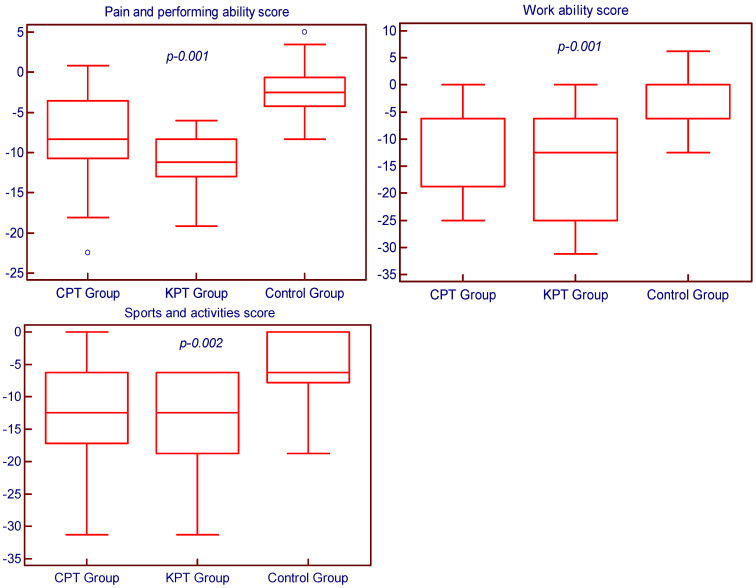
Inter–group comparative analysis of the DASH Outcome Questionnaire score.

**Table 1 medicina-58-01248-t001:** Participants characteristics of age, height, weight and BMI.

Participants Characteristics	CPT Group Mean ± SD	KPT Group Mean ± SD	Control Group Mean ± SD	*p* Value
Age (years)	56.32 ± 6.38	57.11 ± 6.94	55.41 ± 7.22	0.761
Height (cm)	171.32 ± 8.37	172.79 ± 7.26	169.24 ± 7.89	0.404
Weight (kg)	73.89 ± 14.70	75.68 ± 11.75	77.12 ± 11.02	0.746
BMI (kg/m^2^)	25.04 ± 3.47	25.27 ± 2.84	26.93 ± 3.47	0.181

Abbreviations: CPT, classical proprioceptive training; KPT, Kinesimeter proprioceptive training; BMI, body mass index.

**Table 2 medicina-58-01248-t002:** Participants clinical information, sex and manual dominance.

Participants Characteristics	CPT Group		KPT Group		Control Group	
	No.	%	No.	%	No.	%
Sex (female/male)	9/10	47.37/52.63	11/8	57.89/42.11	8/9	47.06/52.94
Hand dominance (left/right)	2/17	10.53/89.47	1/18	5.26/94.74	2/15	11.76/88.24
SLAP tear	4	21.05	5	26.32	3	17.65
SIS	9	47.37	6	31.58	7	41.18
RC tear	6	31.58	8	42.11	7	41.18

**Table 3 medicina-58-01248-t003:** Intra–group comparative analysis of the shoulder JPS.

		CPT Group Mean ± SD	KPT Group Mean ± SD	Control Group Mean ± SD
Flexion (60°)	Initial assessment results	5.93 ± 3.19	6.09 ± 3.73	5.53 ± 2.54
	Final assessment results	4.51 ± 2.60	4.30 ± 3.21	5.06 ± 2.40
	*p* Value	<0.001	<0.001	0.047
Extension (25°)	Initial assessment results	6.46 ± 3.30	6.79 ± 3.63	6.55 ± 3.36
	Final assessment results	5.14 ± 2.45	5.21 ± 2.37	6.37 ± 3.00
	*p* Value	<0.001	<0.001	0.275
Abduction (60°)	Initial assessment results	6.04 ± 3.34	6.35 ± 3.88	6.12 ± 3.10
	Final assessment results	4.51 ± 2.31	4.12 ± 2.59	5.77 ± 2.95
	*p* Value	<0.001	<0.001	0.027
Internal rotation (35°)	Initial assessment results	6.60 ± 3.58	6.98 ± 3.91	6.41 ± 2.77
	Final assessment results	5.19 ± 3.08	5.33 ± 3.42	6.14 ± 2.99
	*p* Value	<0.001	<0.001	0.122
External rotation (35°)	Initial assessment results	6.88 ± 3.32	6.81 ± 3.46	6.67 ± 3.20
	Final assessment results	5.67 ± 2.50	5.44 ± 2.74	6.28 ± 2.92
	*p* Value	<0.001	<0.001	0.203

**Table 4 medicina-58-01248-t004:** Intra–group comparative analysis of the DASH Outcome Questionnaire.

		CPT Group Mean ± SD	KPT Group Mean ± SD	Control Group Mean ± SD
Pain and performing ability score	Initial assessment results	29.52 ± 10.87	30.93 ± 7.62	27.15 ± 6.25
	Final assessment results	21.60 ± 9.53	19.99 ± 7.82	24.94 ± 7.70
	*p* Value	<0.001	<0.001	0.014
Work ability score	Initial assessment results	41.45 ± 16.03	45.07 ± 10.74	42.66 ± 19.69
	Final assessment results	30.59 ± 17.90	29.28 ± 15.87	38.60 ± 19.04
	*p* Value	<0.001	<0.001	0.002
Sports and activities score	Initial assessment results	47.04 ± 14.18	49.34 ± 10.60	50.00 ± 14.32
	Final assessment results	34.87 ± 15.21	34.87 ± 14.78	44.12 ± 13.71
	*p* Value	<0.001	<0.001	0.001

## Data Availability

The data presented in this study are available on request from the corresponding author.

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
