# Peer review of "Effects of a Novel Proprioceptive Rehabilitation Device on Shoulder Joint Position Sense, Pain and Function"

_medicina, 2022, doi:10.3390/medicina58091248_

Round 1

Reviewer 1 Report

The ms reports a study on the effectivity of a new technical device to monitor and train shoulder proproprioception. Applied to three shoulder disorders receiving three types of intervention/therapy combinations, the study shows that the device is as effective as traditional physical therapy. Accuracy of shoulder angle reproduction as well as functional clinical scale of symptoms and pain are part of the patient evaluation. 

The introduction and method sections are sound and well formulated. The results section can be improved substantially by following the advice formulated below. The claims and conclusions can be improved as regards their fit with the findings. Overall this ms almost meets the criteria for publication in the intended special issue of Medicina and I recommend a moderate revision. 

Detailed comments

Lines 54-55: I am not sure whether this is correct. Even when adopting a static posture, muscles have a specific length, and tpart from length changes due to movement, muscle length is also signalled to the brain, right? Proprioception thus involves body perception both in static positions and during motion. 

Line 58: Replace "The experiment bij Kelso et al. demonstrated" by "Kelso et al. (1980) demonstrated"

Line 215-216: Was this 1D, 2D, 3D angular motion...?

Lines 244-246: Reformulate since the difference between angles and degrees of the shoulder joint is unclear. Lines 254-256: The difference....was measured (it is not plural).

 Line 257: Were the different shoulder disorders comparable as to the expected errors? I can imagine that systematic errors may reflect resistance to move or reproducing angles close to the minimum or maximum joint angle which are known to induce pain. Why were absolute errors measured rather than signed errors?

Line 318: The results as described in the running text are presented in terms of presence or absence of differences together with relevant p-values. For the size and direction of the effects the reader needs to study the tables. Highlighting important results in terms of size, direction and p-values preferably in relation to control aspects of the design or the hyptheses, is recommended. The results section thus can benefit substantially from rewriting according to a more conventional results reporting strategy.

Lines 319-321: Post-intervention results were LOWER than the baseline assessment results...but the authors should remind the reader of the relevant  dependent measure, which is a reproduction/proprioceptive error. Then it becomes clear that, as expected, proprioception was better/improved/'higher' following the interventions, rather than 'lower' (which means that the errors were smaller). Disambiguating the reporting in the results section will benefit this paper.    

Line 335: Why reporting medians along means....? Were means of medians entered in the ANOVAs? I advise the authors to report economically, i.e. either the means of the medians. SDs are of course relevant information. 

Line 337: grammatical error: encountered. The discussion section should be upgraded with respect to language use.

Line 381: replace "and" by "an"

Line 384: remove "a" before feedback

Line 385: remove "a" before real-time

Line 409: add to this conclusion: 'like conventional physiotherapy' 

Author Response

Esteemed Editor and Reviewer from Medicina Journal,

Thank you for giving us the opportunity to submit a revised draft of the manuscript. Please see the attached point-by-point response to Reviewer #1 suggestions and comments.

Alexandra – Camelia Gliga

Author Response

Esteemed Editor and Reviewer from Medicina Journal,

Thank you for giving us the opportunity to submit a revised draft of the manuscript. Please see the attached point-by-point response to Reviewer #2 suggestions and comments.

Alexandra – Camelia Gliga
